

# Data-driven sales optimization with regression and chaotic pattern search

Sandhya Rani Gaddam[1], Sarada Jayan[1], Pentakota Ravi[2] and Bilal Alatas[3]

[1] Department of Mathematics, Amrita School of Engineering, Amrita Vishwa Vidyapeetham, Bengaluru, Karnataka, India
[2] Operations Department, Genpact, Bengaluru, Karnataka, India
[3] Department of Software Engineering, Firat University, Elzig, Turkey

## ABSTRACT

Lead generation is the process of gaining potential customers' interest to increase future sales, and it is an essential part of many businesses' (amusement parks, theme parks, clubs, *etc.*) sales processes as their membership is more expensive. The main objective of these businesses is to increase the count of customers. By generating sales leads, a club/park can find leads who have already expressed interest in its products and services and access their audience potential, allowing them to focus on future marketing and sales efforts on those leads that are more likely to convert. The current work focuses on how to convert a lead to a customer in optimum number of days. We collect two kinds of data: customer data and lead generation data. The customer data consists of all the leads who have taken the membership, and the lead generation data consists of all current leads. The details of those converted from a lead into a customer in the last 60 days are filtered out from the customer data. Using this data, patterns are generated, which are used to predict the following activity (step) for qualified leads, along with the optimal number of days required to complete that activity. This optimal number of days is found using the Hybrid Chaotic Pattern Search Algorithm (HCPSA). This novel approach here helps in boosting sales by prioritizing leads who have expressed interest and identifying the optimal window for converting them into paying customers. This strategy holds significant potential to benefit businesses across various industries.

## INTRODUCTION

In the marketing field, a 'lead' is a person/organization who has shown interest in a company product and a 'customer' is a person who has brought the product/membership of that company. 'Velocity to sale' is the number of days needed to convert a lead to a customer. In B2B and B2C companies (*Huttelmaier & Heigl, 2023*; *Desai & Vidyapeeth, 2019*), lead generation is essential since their products are often expensive, and website visitors are less likely to purchase directly from them. Businesses must generate leads since gathering a prospective customer's contact information allows them to market to them later, even if the customer does not purchase immediately. Most leads are obtained through referrals from existing customers or as a direct result of advertising and publicity.

Corresponding author
Sarada Jayan,
j_sarada@blr.amrita.edu

Converting leads to customers is the most challenging task for the sales/marketing team. To transform leads into customers, the sales and marketing team must focus on various strategies (*Kiumarsi et al., 2014*), such as sending regular follow-ups, offering discounts by email and calls, asking for referrals, *etc*. Lead generation happens through all the digital channels (both free and paid) at the company's disposal. These channels include social media, search engine rankings, emails, display advertisements, and the company's blog/website. A company's marketing strategy aims to initially attract and later convert an interested person to a lead for salespeople *via* the website and supporting digital channels (*Grublješič & Čampa, 2016*). Recent studies show that only 10% to 15% of leads reach the bottom of the sales funnel (*Grublješič & Čampa, 2016*) (Fig. 1).

Once the leads are generated, the sales and marketing team must qualify them based on their interest. Qualifying leads (*Wu, Andreev & Benyoucef, 2023*) helps them determine which leads are worth talking to and which should be ignored. Using the lead qualification process (*Grublješič & Čampa, 2016*; *Wu, Andreev & Benyoucef, 2023*) that gives scores to leads, the sales and marketing team gets visibility on leads to pick up the qualified leads (the leads that have a high probability of converting into customers). Afterwards, they find strategies to convert qualified leads into customers to reach their goals quickly. They should concentrate on some measures like conversion rate, velocity to sales, *etc*.

Existing literature offers limited guidance on converting leads into customers. The motivation for this article is to explore strategies for businesses to convert leads and boost their sales. The current work focuses on converting qualified leads into customers within optimum number of days. Two kinds of data are collected—customer data and lead generation data. The customer data is the historical data that consists of the processes through which leads were converted to customers, and the lead generation data consists of a list of current leads. Customers who got converted in the last 60 days are filtered from the customer data. Patterns are generated, and using this filtered data and these patterns, we:

 (i)  predict the next step for qualified leads;
(ii)  predict the number of days to complete this step.

The optimal number of days to complete the activity by a lead can be found using any optimization algorithms (*Feng et al., 2017*; *Fister et al., 2015*; *Rani, Jayan & Alatas, 2023*; *Hooke & Jeeves, 1961*; *Rani & Jayan, 2021*; *Deb, 1995*). The optimization algorithm used in this work is the Hybrid Chaotic Pattern Search Algorithm (HCPSA) (*Rani, Jayan & Alatas, 2023*) as this is one of the best algorithms to get the global solution. This hybrid algorithm has two stages. In the first stage, chaotic maps are used to obtain a solution near the global solution. Stage two uses a local pattern search method with the initial point as the solution from stage one to obtain the exact global optimum point.

The novelty of this work is that, for the first time an optimization algorithm with machine learning tools is used to give strategies for marketing team and to optimize the Velocity to Sales.

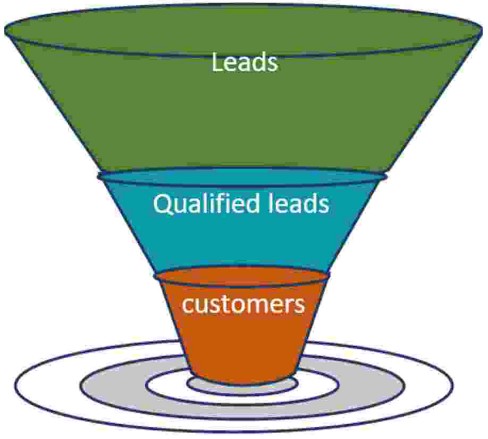

**Figure 1 Sales funnel.**

"Preliminaries" in this article describes HCPSA and the chaotic map used in it. Velocity to sales is discussed in "Velocity to Sales (VTS)". The details and procedure to optimize velocity to sales are discussed in "Optimizing Velocity to Sales", and finally, the article is concluded in "Conclusions".

## PRELIMINARIES

### Chaotic maps

Chaotic maps generate chaotic numbers possessing attributes such as pseudo-randomness, sensitivity, regularity, and ergodicity, thereby enhancing the Chaos Optimization Algorithm (COA). These properties of chaotic numbers aids COAs in systematically exploring the entire search space over time. Several chaotic maps (*Feng et al., 2017*; *Fister et al., 2015*; *Rani, Jayan & Alatas, 2023*; *Rani & Jayan, 2021*; *Vallabhaneni et al., 2020*; *Varol Altay & Alatas, 2020*; *Bingol & Alatas, 2023*) are defined in the literature like logistic map, Chebyshev map, sine map, circle map, ICMIC map, tent map, neuron map, Kent map, Gauss map, *etc.*

Chebyshev map: Generally, Chebyshev chaotic maps are used to deal with security issues, digital communication, neural networks and in global optimization algorithms. The iterative formula that generates chaotic sequences using Chebyshev map is as follows.

$$x_{n+1} = cos(\alpha\ cos^{-1}(x_n)),\ x_n \in (-1, 1),\ \alpha \in \mathbf{R}.$$

Based on the Lyapunov exponent and scatter diagram, we choose the best value for the $\alpha$ as 8 as explained in *Rani, Jayan & Alatas (2023)*. Here the sequence $x_n$ is the set of chaotic numbers scattered in the interval $(-1, 1)$. The fixed point $x^*$ of a chaotic map $x_{n+1} = f(x_n)$ is a point at which $x^* = f(x^*)$. The chaotic nature of a map will get lost at this point and so care must be taken to avoid the usage of this point as an initial point in any algorithms where chaotic maps are used. The fixed points of Chebyshev map are $-0.9397, -0.9010, -0.5, -0.2225, 0.1736, 0.6235, 0.7660$ and $1$.

## Hybrid chaotic pattern search algorithm (HCPSA) (*Rani, Jayan & Alatas, 2023*)

COAs typically excel at swiftly navigating towards the vicinity of the optimum, yet they may require additional computations to precisely reach the optimum from that neighborhood. To mitigate this, hybrid algorithms like HCPSA combine COA in the initial stage to approach the function's minimum neighborhood and employ a local search algorithm in the subsequent stage to efficiently attain the minimum, as detailed in reference (*Rani, Jayan & Alatas, 2023*). *Rani, Jayan & Alatas (2023)* describes an algorithm well-suited for optimizing functions with multiple variables. This algorithm is particularly effective for higher dimensional problems. For readers' convenience, a description of the HCPSA algorithm from *Rani, Jayan & Alatas (2023)* is provided in this Section.

Stage I:

Step 1: Input the objective function $f$, the lower and upper bounds for the variable as $L$ and $U$. Choose the number of iterations for stage I as $N$. Set $k = 1$, initialize the dimension i and choose $fmin = +\infty$.

Step 2: Initialize the initial value of the chaotic variable $c^k$. This initial value is same for all dim in order to reduce the functional evaluations and computational time. The initial value should be selected from the defined interval for each of the chaotic maps and should not be any of the fixed points of the respective chaotic maps.

Step 3: Map the chaotic variable $c^k$ into the optimization variable $X_i^{(k)}$ $i = 1, 2, \ldots, n$ by using the following equation. For Chebyshev map we use $X_i^{(k)} = \left(\frac{U+L}{2}\right) + \left(\frac{U-L}{2}\right)c^k$ $i = 1, 2, \ldots, n$ (This transformation is done since the Chebyshev map generates pseudo random numbers between −1 and +1, but we need search points between L and U.)

Step 4: Compute the function value $f\left(X_i^{(k)}\right)$.

If $f\left(X_i^{(k)}\right) < fmin$ then $fmin = f\left(X_i^{(k)}\right)$ and the optimal solution is $Xmin = X_i^{(k)}$, $i = 1, 2, \ldots, n$

Step 5: Generate the next chaotic variable $c^{k+1}$ by applying the chaotic map.

Step 6: $k = k + 1$ If $k \leq N$ go to Step 3, else go to Step 7.

The output of stage I is *Xmin*, which is in the neighborhood of global optima.

Stage II:

Step 7: Choose *Xmin* as the initial point of stage II.

Step 8: Use Pattern Search Algorithm to find global optima (*Rani, Jayan & Alatas, 2023*; *Hooke & Jeeves, 1961*).

## Ordinary least square method of regression

Ordinary Least Squares (OLS) regression is a fundamental tool used in statistics and machine learning to understand the relationship between variables. It is particularly useful for linear regression models, to find a straight line that best fits the data points. OLS works by minimizing the squared differences between the actual values of a dependent variable

and the values predicted by the equation based on independent variables. In this article OLS regression method is used to obtain the weights used in the velocity to sales function.

## VELOCITY TO SALES

For B2B and B2C companies, it is essential to understand and work on velocity to sales to improve their marketing strategies that can increase sales. In the current work, velocity to sales (the number of days needed to convert a lead to a customer) is explained with an example of a club that aims to get more memberships. The club encompasses a variety of parks, resorts, and spas, as well as beaches, islands, and villas. Membership in this club will give a 50% discount on accommodation in the club's various parks, villas, and resorts and on availing the services of their spas. Also, members will have flexibility in deciding the time, venue, and frequency of their vacation trips.

The main objective of this club is to increase the number of customers who takes their club's membership. In order to increase this, the marketing and sales team must do some activities, which are listed below.

- Need to generate many leads.
- Generated leads need to be given scores based on their interest.
- Based on the lead scores, qualified leads need to be identified.
- Try to convert qualified leads into customers.

Our proposed work helps the marketing and sales team to easily convert a qualified lead to a customer in fewer days by optimizing velocity to sales.

## OPTIMIZING VELOCITY TO SALES

The proposed model is trained using a sample of 20,000 customer data (with group IDs A1-A20000) who have converted from a lead to a customer in the last 60 days and tested on lead data. The customer data contains information about customers who have become club members. This data consists of six columns, namely customer ID, customer ID Type (Lead/Member), Category (Marketing/Sales teams), Activity (sent emails/contacted through a call/personal meetings), Date of the Activity, Date when membership was taken, and Opportunity Flag. If the customer shows interest on a specific date, the opportunity flag is displayed as one; otherwise, it is displayed as zero. Table 1 gives the history of a customer with customer ID A1. Details of all the activities done by the marketing/sales team to convert A1 from lead to a member can be seen in this table. The sales team called customer A1 seven times and had an in-person conversation three times. The marketing team had sent multiple emails till December 2022. Finally, after a call from the sales team, customer A1 took the membership on 17th January 2022. The activities of a customer after taking a membership will also be recorded, but these details are not required for optimizing the Velocity to Sales (VTS).

Lead data contains information about leads who have expressed interest in the club's products and services but have not yet taken membership. Lead data consists of five columns, the same as the first five columns in the customer data. Our goal is to convert these leads into members by forecasting the ideal activities that need to be taken by the

**Table 1 History of customer ID A1 (Member).**

| Customer ID | Customer ID type | Category | Activity | Date of activity | Membership date | Opportunity flag |
|---|---|---|---|---|---|---|
| A1 | Lead | Sales | Outbound call | 11/19/2022 | 1/17/2022 | 0 |
| A1 | Lead | Marketing | Email | 11/20/2022 | 1/17/2022 | 0 |
| A1 | Lead | Marketing | Online | 11/23/2022 | 1/17/2022 | 0 |
| A1 | Lead | Sales | Outbound call | 11/23/2022 | 1/17/2022 | 0 |
| A1 | Lead | Marketing | Email | 11/24/2022 | 1/17/2022 | 0 |
| A1 | Lead | Sales | Outbound call | 11/25/2022 | 1/17/2022 | 0 |
| A1 | Lead | Sales | Outbound call | 11/25/2022 | 1/17/2022 | 0 |
| A1 | Lead | Marketing | Email | 11/27/2022 | 1/17/2022 | 0 |
| A1 | Lead | Marketing | Email | 11/29/2022 | 1/17/2022 | 0 |
| A1 | Lead | Marketing | Email | 12/01/2022 | 1/17/2022 | 0 |
| A1 | Lead | Marketing | Email | 12/06/2022 | 1/17/2022 | 0 |
| A1 | Lead | Sales | Email | 12/08/2022 | 1/17/2022 | 0 |
| A1 | Lead | Sales | In-person conversation | 12/10/2022 | 1/17/2022 | 1 |
| A1 | Lead | Sales | In-person conversation | 12/11/2022 | 1/17/2022 | 0 |
| A1 | Lead | Sales | In-person conversation | 12/12/2022 | 1/17/2022 | 0 |
| A1 | Lead | Sales | Outbound call | 12/19/2022 | 1/17/2022 | 0 |
| A1 | Lead | Marketing | Email | 12/26/2022 | 1/17/2022 | 0 |
| A1 | Lead | Sales | Outbound call | 1/17/2022 | 1/17/2022 | 0 |
| A1 | Member | Sales | Outbound call | 1/17/2022 | 1/17/2022 | 1 |

marketing/sales team and predicting the optimum number of days required for performing the next activity. "Predicting" this will help the marketing and sales teams to increase sales. "Prediction of next activity for a lead" explains the model that will use the customer data as the trained data and predict the following activity for the leads in the lead generation data (test data). "Prediction of Optimal Number of Days for Each Activity" gives a model that will evaluate the optimum number of days required to complete the predicted activity given in "Prediction of next activity for a lead".

Figure 2 illustrates the flowchart of the proposed model, which is elucidated as follows. The customer data is initially cleaned to ensure accuracy and completeness. Rolling up data helps to aggregate the data for easier analysis. After processing the customer data, the model extracts patterns. Like customer data, the lead data is cleaned and summarized. The model then uses the patterns learned from the customer data to identify promising leads within the lead generation data set.

## Prediction of next activity for a lead

The first and foremost step is to verify that the data is in ascending order based on the date column; otherwise, it needs to be arranged. The activities for customer A1 in date-wise order are provided as an example below. It can be noted that the consecutively repeated activities are removed from this data.

outboundcall email online outboundcall email outboundcall email inpersonconversation outboundcall email outboundcall.
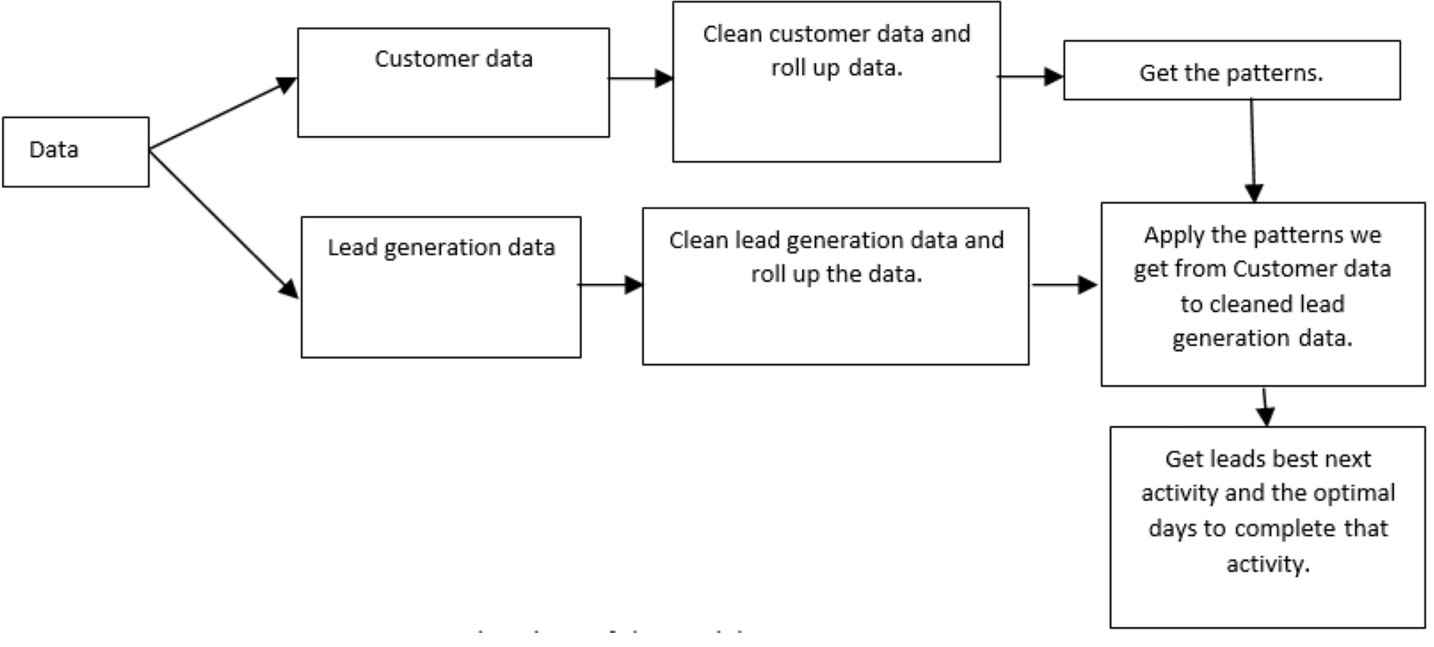

**Figure 2 Flowchart of the model.**

This sequence is cleaned to get a good prediction by removing repeated activities. The repeated activity that is retained is the latest one. After cleaning the customer data of A1, we obtain:

online inpersonconversation email | outboundcall.

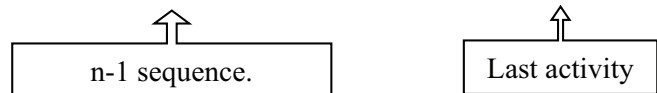

The above sequence is divided into two parts: the first n-1 activities shall be treated as the independent variables, and the last activity is considered the dependent variable. By this, the model shall be a supervised learning model.

After following this process for customers A1–A5, the cleaned data is depicted in Table 2.

All possible combinations of the Customer clean sequence of all customers with a minimum length of three are identified. For customer ID A1, all such combinations are listed in column 2 of Table 3.

After following this process for 20,000 customers, a frequency table can be generated based on the number of times the combination appears. Table 4 provides this frequency count for the training data that is used.

As per the given data, the probability is high for a person to take membership if an in-person conversation or an outbound call is made after direct mail and email. Based on Table 4, a suggestion of activities to be followed after a particular sequence in descending order of probability is given in Table 5.

**Table 2  Cleaned data of customers.**

| Customer ID | Cleaned customer sequence | Sequence-up-to n-1 | Sequence last word |
|---|---|---|---|
| A1 | Online inpersonconversation email outboundcall. | Online inpersonconversation email | Outboundcall |
| A2 | Inroomletter outboundcall email inpersonconversation | Inroomletter outboundcall email | Inpersonconversation |
| A3 | Online webpage email chat outboundcall | Online webpage email chat | Outboundcall |
| A4 | Socialmedia online chat directmail outboundcall email | Socialmedia online chat directmail outboundcall | Email |
| A5 | Directmail online webpage chat evt email inroomletter inpersonconversation outboundcall | Directmail online webpage chat evt email inroomletter inpersonconversation | Outboundcall |

**Table 3  All combinations of the customer-clean-sequence of A1.**

| Customer ID | Combinations for cleaned customer sequence | Sequence-up-to n-1 | Sequence last word |
|---|---|---|---|
| A1 | Online inpersonconversation email outboundcall. | Online inpersonconversation email | Outboundcall |
| A1 | Online inpersonconversation email | Online inpersonconversation | Email |
| A1 | Inpersonconversation email outboundcall. | Inpersonconversation email | Outboundcall |

**Table 4  Frequency count.**

| Customer-clean-sequence all combinations | Sequence-up-to n-1 | Sequence last word | Count |
|---|---|---|---|
| Directmail email inpersonconversation | Directmail email | Inpersonconversation | 526 |
| Directmail email outboundcall | Directmail email | Outboundcall | 499 |
| Directmail email webpage | Directmail email | Webpage | 100 |
| Directmail email chat | Directmail email | Chat | 85 |
| Directmail email call | Directmail email | Call | 39 |
| Directmail email webcast | Directmail email | Webcast | 31 |
| Directmail email inroomletter | Directmail email | Inroomletter | 26 |
| Directmail email online | Directmail email | Online | 23 |
| Directmail email evt | Directmail email | Evt | 17 |

**Table 5  Predicted activities based on frequency.**

| Sequence-up-to n-1 | Predicted activities |
|---|---|
| Directmail email | Inpersonconversation \| outboundcall \| webpage |
| Call email | Outboundcall \| inpersonconversation \| chat |
| Directmail call | Outboundcall \| email \| inpersonconversation |
| Online directmail webcast | Email \| inpersonconversation \| outboundcall |
| Online email | Directmail \| outboundcall \| webpage |
| Online email chat | Outboundcall \| inpersonconversation \| direct |
| … | … |

After direct mail email, the sales/marketing team may go for an in-person conversation, an outbound call, or a webpage because they are the topmost options.

Using the probability obtained using the frequency count (as shown in Table 4), a prediction of the next activity is made for the sequences in column 3 of Table 2. If the predicted activity is the same as the one in column 4 of Table 2, the result is taken as 'TRUE'; otherwise, it is considered as 'FALSE'. Table 6 shows this step. Based on this Tue/False column, the accuracy of the model is calculated and is found to be 85% for the data taken.

Next, the data pertaining to the 'FALSE' cases are removed from the training set. Using the new training set, a model is created that will predict the next activity for every individual in the lead data set (testing set).

## Prediction of optimal number of days for each activity

The previous section provides the sequence of the activities that must be followed based on the trained data. This section deals with deciding the optimal number of days to be spent for each activity, thus minimizing the velocity of sales.

Velocity of sales is $VTS = x_1 + x_2 + \ldots + x_n$, where $x_i$ is the number of days taken to complete $i^{th}$ activity. Each activity carries a different weightage. Thus in order to find the optimal $x_i$ for each activity, we need to

$$\text{Minimize } w_1 x_1 + w_2 x_2 + \ldots + w_n x_n \tag{1}$$

where $w_1, w_2, \ldots, w_n$ are the weights corresponding to the paths $x_1, x_2, \ldots, x_n$ respectively. The procedure for minimization of the given function is explained below:

Step 1: Bifurcate the group IDs whose sequence starts with a particular activity. For example, all the sequences starting with the activity "online", all the sequences starting with the activity "e-mail", all the sequences starting with the activity "outbound call", *etc.*, are separated and considered as different buckets.

The following is an example of a bucket with different group IDs, all of which are starting online.

A1—online inpersonconversation email outboundcall.

A2—online webpage email chat outboundcall.

A3—online webpage chat email inroomletter inpersonconversation outboundcall.

A4—online inroomletter outboundcall email inpersonconversation.

…

Step 2: Tabulate each path and the number of days required to complete each activity in these paths for each customer ID in each bucket, as provided in Table 7. The total number of days taken to complete all activities of each customer is also tabulated. An example of this for a bucket in which all paths start with 'online' is provided in Table 7.

Step 3: Pivot the table obtained in the previous step by keeping group IDs as rows and paths as columns so that it becomes appropriate for applying machine learning models. After pivoting Table 7, we get Table 8.

**Table 6 Checking the accuracy of the predicted activity.**

| Sequence up to n-1 | Last word of the sequence | Predicted last word (activity) of the sequence | True/False |
|---|---|---|---|
| Online Inpersonconversation Email | Outbound call | Outbound call \| chat \| call | True |
| In room letter Outbound call Email | In person conversation | In person conversation \| direct mail \| webpage | True |
| Online Webpage Email chat | Outbound call | Direct mail \| in person conversation \| webpage | False |
| Email Direct mail | Outbound call | Out bound call \| in person conversation \| webpage | True |
| Outbound call Direct mail | Email | Email \| in-person conversation \| webpage | True |

**Table 7 Number of days taken to complete the activities.**

| Customer ID | Path | Days to complete next activity | Path starts with | Number of days taken to complete all activities of each customer |
|---|---|---|---|---|
| A1 | Online-inpersonconversation | 20 | Online | 58 |
| A1 | Inpersonconversation-email | 15 | Online | |
| A1 | Email-outboundcall | 23 | Online | |
| A2 | Online-webpage | 5 | Online | 44 |
| A2 | Webpage-email | 3 | Online | |
| A2 | Email-chat | 15 | Online | |
| A2 | Chat-outboundcall | 21 | Online | |
| … | … | | | |

**Table 8 Number of days to complete each activity.**

| Customer ID | Number of days | Days to complete online-inpersonconversation | Days to complete inpersonconversation-email | Days to complete email-outboundcall | Days to complete online-webpage | Days to complete webpage-email | Days to complete email-chat | Days to complete Chat-outboundcall | …….. |
|---|---|---|---|---|---|---|---|---|---|
| A1 | 58 | 20 | 0 | 0 | 0 | 0 | 0 | 0 | |
| A1 | 58 | 0 | 15 | 0 | 0 | 0 | 0 | 0 | |
| A1 | 58 | 0 | 0 | 23 | 0 | 0 | 0 | 0 | |
| A2 | 44 | 0 | 0 | 0 | 5 | 0 | 0 | 0 | |
| A2 | 44 | 0 | 0 | 0 | 0 | 3 | 0 | 0 | |
| A2 | 44 | 0 | 0 | 0 | 0 | 0 | 15 | 0 | |
| A2 | 44 | 0 | 0 | 0 | 0 | 0 | 0 | 21 | |
| … | | | | | | | | | |

Step 4: To the pivoted table of Step 3, apply the ordinary least square (OLS) model of regression (*Daliya, Ramesh & Ko, 2021*; *Sathyadevan & Chaitra, 2015*) by taking the number of days (second column of Table 8) as dependent and all paths (remaining columns of Table 8) as independent variables. The result of the OLS model gives an approximation for the coefficients of $x_i$'s, *i.e.*, $w_i$'s in the objective function $w_1 x_1 + w_2 x_2 + \ldots + w_n x_n$ along with a standard error of regression, $\varepsilon$. For better accuracy of the proposed model, we consider $w_i$'s also as unknowns with bounds as the coefficient of $x_i$ obtained using the OLS $\pm \varepsilon$ for $i = 1, 2, \ldots, n$.

The minimum and median number of days corresponding to each path $x_i$ in a bucket are taken as the lower and upper bounds for each $x_i$.

Step 5: Apply HCPSA to minimize $w_1 x_1 + w_2 x_2 + \ldots + w_n x_n$ with 2n unknowns, each with bounds, as given in Step 4. The solution of HCPSA gives optimal $x_i$'s and $\sum x_i$ will give the optimal velocity of sales for one particular bucket.

Step 6: Repeat Steps 2 to 5 for all the buckets and consolidate all bucket outputs in a single table. Remove the duplicates, if any. Finally, we obtain a list of distinct paths along with the optimal number of days required to complete the activity in that path.

The model in "Prediction of next activity for a lead" trains the customer data to predict the next activity for a lead and, the model in "Prediction of optimal number of days for each activity" suggests the number of days required to complete this activity based on the trained customer data. The results thus obtained from the training data are then used to decide the following activity for leads in the lead data (testing data) and to set a target (number of days to complete the prescribed activity) for the salespeople, which, if achieved shall optimize the velocity to sales.

The code for this research was written in Python for easy replicability. It was developed and run on a standard laptop with 8 GB RAM and an Intel i5 processor.

Executing this work presented few key challenges that includes data cleaning difficulties and arranging the data in the required sequence. By implementing improved data processing techniques, we can significantly enhance the proposed algorithm's performance. This highlights the potential for further research and development.

## CONCLUSIONS

Many businesses strive hard to enhance their sales performance. However, achieving their sales targets necessitates adherence to specific processes by the marketing and sales team, which includes generating a substantial number of leads. Based on the interest of the generated leads, qualified leads are filtered, and the marketing/sales teams will focus on how to convert these qualified leads into customers. The proposed work provides techniques to convert qualified leads into customers based on the historical data of existing customers. This data consists of the details of the marketing strategies that helped in converting leads to customers. Based on a study of probability on this data, the following activity for a lead is predicted and using regression on this data coupled with a multivariable optimization algorithm (HCPSA), the optimal number of days required to complete the predicted activity is obtained. These results can help the sales team to decide on the course of action to convert qualified leads into customers. By updating customer

data and lead information weekly, the efficiency of this process is guaranteed. Moreover, this effort serves as a readily deployable tool for augmenting sales.

### Funding
The authors received no funding for this work.

### Competing Interests
Bilal Alatas is an Academic Editor for Peer J. Pentakota Ravi is employed by Genpact.

### Author Contributions
- Sandhya Rani Gaddam conceived and designed the experiments, performed the experiments, analyzed the data, performed the computation work, prepared figures and/or tables, authored or reviewed drafts of the article, and approved the final draft.
- Sarada Jayan conceived and designed the experiments, performed the experiments, analyzed the data, performed the computation work, prepared figures and/or tables, authored or reviewed drafts of the article, and approved the final draft.
- Pentakota Ravi conceived and designed the experiments, performed the experiments, analyzed the data, performed the computation work, prepared figures and/or tables, authored or reviewed drafts of the article, and approved the final draft.
- Bilal Alatas conceived and designed the experiments, performed the experiments, analyzed the data, performed the computation work, prepared figures and/or tables, authored or reviewed drafts of the article, and approved the final draft.

### Data Availability
The article is a theoretical work and so does not have raw data.

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
