# Peer review of "Data-driven sales optimization with regression and chaotic pattern search"

_PeerJ Computer Science, doi:10.7717/peerj-cs.2144_

## Round 0.1 · original submission · Major Revisions

Kindly address the reviewers comments and improve your paper quality. (Presentation and English editing is also recommended for professional write up)

**Language Note:** The Academic Editor has identified that the English language must be improved. PeerJ can provide language editing services - please contact us at [email protected] for pricing (be sure to provide your manuscript number and title). Alternatively, you should make your own arrangements to improve the language quality and provide details in your response letter. – PeerJ Staff

Reviewer 1 ·

Basic reporting

The paper gives an algorithm for optimizing lead conversion for businesses with expensive memberships, like amusement parks and clubs. Authors have highlighted the importance of focusing efforts on leads who have already shown interest and finding the ideal timeframe for converting them into paying customers. This is a novel idea that can help many business firms. By following the methodology in this paper a business company can use their historic data to:
(i) Develop strategies to nurture leads and move them through the conversion funnel faster, by predicting the next step for the lead at each stage
(ii) Predict the optimal number of days for conversion from lead to customer using data analysis and an optimization algorithm.

With these, the methodology given here becomes a “Lead-to-Customer Conversion Tool” that can be used for improving sales. This tool can attract a lot of citations for this paper. But clarity is needed for the below points:

1. In Section 2 where key concepts are introduced, the algorithm (2.2) mentions about chaotic maps and fixed points of the chaotic maps. Which all chaotic maps can be used for the purpose? What are fixed points of chaotic maps?
2. The main motivation of this article needs to be clarified in the introduction section to make it easier for future readers. Then, the main idea and how the given problem was addressed can be found.
3. Why OLS method of regression is chosen?
4. Section 2 needs to be restructured for better clarity. Specifically, focus on explaining the OLS regression method, chaotic maps, and their essential features.
5. How did the chaotic element improve the performance of the search process?
6. The HCPSA algorithm that is used for optimizing the number of days for each activity, has been developed by the same authors in a different paper. A mention of that is essential in introduction as well as in Section 2.2 where the description of this algorithm is provided.
7. Despite the existence of well-established optimization algorithms such as particle swarm optimization, why HCPSA is selected for optimizing velocity to sales?

Experimental design

8. Running environment according to the software and hardware should be written in “4. Optimizing Velocity to Sales” section.
9. Idea of embedding chaos theory into optimization algorithms is used in some prestigious studies. About integrating chaos and chaotic maps, more works may be analysed for mentioning the methodological novelty of this study in Introduction section.
10. The article does not give any information on the difficulties faced during the execution of the methodology or about any disadvantages or shortcomings of it. Is the methodology restricted to any particular type of data? More information about data and pre-processing should be given.

Validity of the findings

11. Some critical and theoretical discussions should be added to the discussion of results to convince the reader of the proposed method and how it can be better than other methods in the field.
12. In Section 2.2, steps of the algorithm should be written according to a single style. For example, see: “Step1”, “Step 5”. Usage of blank character is important.
13. Equations should be used with equation numbers with related references. Definitions of the variables and the boundaries should also be defined.
14. Variables should be written in italic similar to the usage in the equations.
15. Blank character should be properly used in the article. “many businesses'(amusement parks”, “…following activity(step) for…”, “…a customer. In B2B and…”, “…Algorithm (HCPSA)[8]…”, “chaotic maps [6-8,10,12-14] are”, “…global optima[8,9].”, “…of regression [15,16]…”, etc. should be corrected.

Additional comments

16. All references should be listed according to the PeerJ referencing style. See for example Reference [8], [10], and [12].
17. “(second column of table 8)” should be corrected as “(second column of Table 8)”.
18. “remaining columns of table 8” should be corrected as “remaining columns of Table 8”.

Cite this review as

Reviewer 2 ·

Basic reporting

• Abstract needs to be modified by adding the specific drawbacks of the existing works. Furthermore, quantitative results are not clearly reported in the abstract.
• The novelty of the work needs to be highlighted in the abstract of the paper.
• The paper explains about a methodology to convert qualified leads to customers. More information with regard to selection of qualified leads based on scores needs to be provided in the paper.
• Already there are deep learning techniques in NLP such as LSTM, RNN, etc. to predict the next activity. Then, what is the benefit of using the proposed methodology?
• In section 4.1 (line number 178) it is mentioned that ‘the consecutively repeated activities need to be removed from the data’. (a) What is the strategy followed to remove these repeated activities? (b) Won’t the information be lost when you remove these repetitions?
• Are you applying any clustering techniques for qualifying leads?
• The article lacks insights into potential challenges encountered during the methodology’s implementation and does not discuss any limitations or drawbacks. Are there data-type restrictions associated with this methodology? In order for readers to better understand under which conditions the results should be interpreted, the limitation of the proposed HCPSA (if there is any) should be clarified.
• An elaboration on the data requirements and a more in-depth explanation of the pre-processing procedures need to be given. There is insufficient information on data and data processing for the HCPSA method.
• Chaotic maps have been incorporated into the pattern search algorithm to optimize velocity to sales. What is the need of this selection when powerful evolutionary algorithms are available? The motivation on adapting the HCPSA for the problem of optimization of velocity to sales is absent.
• Encoding type or representation type and computing of fitness function of the HCPSA method for optimizing the velocity to sales are not clearly described in the paper.
• All variables used in the equations should be written in italic. For example, see line numbers 118, 121, 178, 257, in Section 4.1.
• All definitions of variables in the equations should be reviewed in order to further clarify these equations. Their definitions, intervals, etc. should be defined.
• Presented tables need more explanation.
• The authors should pay attention the usage of blank characters. See for example, line numbers 248, 253, etc. The authors should also pay attention for the usage of uppercase. “table”, “step”, “section” etc. should be corrected.
• "Conclusions" section should be expanded considering additional comments on the results with some specific ideas for future works.

Experimental design

no comment

Validity of the findings

no comment

Additional comments

no comment

Cite this review as

·

Basic reporting

.Rewrite title
.Improve the writing style of the paper and write in reasonable sentence length like in the conclusion (They need to generate lots of leads, This work can be used as a ready-made tool to increase sales. )
.Use sentence starters like similar, moreover, therefore according to sentence requirement.

.Write some broad-level descriptions for Stage I and Stage II in section 2.2.

Experimental design

In experimental framework needs to improve the research flowchart and mention which steps were adopted to clean the data and improve the overall structure of the diagram.

Validity of the findings

Write the limitations and strength of the conducted research.

---

## Round 0.2 · Major Revisions

I append my own comments on your revision:

Lack of Novelty in Chaotic Maps:
The use of chaotic maps such as the Chebyshev map is well-documented in literature for various applications. The proposal needs to clarify what new aspects or improvements it brings to the use of these maps. Are there new properties of the maps being utilized? Is there a new application or a significant improvement in performance for existing applications?

Clarity on Hybrid Chaotic Pattern Search Algorithm (HCPSA):
While the proposal mentions the effectiveness of HCPSA in optimizing multi-variable functions, it does not clearly state how this hybrid approach differs significantly from existing algorithms. What makes HCPSA superior or different? Detailing specific performance metrics or scenarios where HCPSA outperforms other methods could strengthen this section.

Integration and Application of OLS in VTS:
The use of Ordinary Least Squares (OLS) regression to analyze velocity to sales introduces a statistical method to a business process optimization problem. However, the proposal could be enhanced by discussing why OLS is chosen over other, potentially more robust methods that could handle non-linear relationships or high variability common in sales data.

Methodological Details:
The proposal could benefit from more detailed methodological descriptions. For instance, how are chaotic variables mapped into optimization variables? What specific improvements or innovations does this mapping provide over traditional methods?

Practical Implications and Limitations:
While the proposal outlines what the research aims to achieve, it lacks a discussion on the practical limitations or challenges that might be encountered. For instance, are there computational limitations, assumptions in the model that may not hold in real-world scenarios, or specific conditions under which the proposed methods may fail?

Risk of Overfitting in VTS Optimization:
The model's training on past customer data to predict sales activities might risk overfitting, where the model learns the training data too well, including the noise and anomalies, which does not generalize well on unseen data. Discussing strategies to mitigate overfitting and ensure model robustness would be beneficial.

Environmental and Ethical Considerations:
If the proposal involves data collection or has potential environmental impacts, it should address these aspects. Ethical considerations, particularly in handling customer data, privacy, and implications of predictive modeling in sales, are crucial.

Reviewer 1 ·

Basic reporting

All changes have been completed.

Experimental design

All changes have been completed.

Validity of the findings

All changes have been completed.

Cite this review as

Reviewer 2 ·

Basic reporting

Author has done required changes; The paper can be accepted in the current forme

Experimental design

no comment

Validity of the findings

no comment

Additional comments

no comment

Cite this review as

---

## Round 0.3 · accepted · Accept

The article is Accepted

·

Basic reporting

no comments

Experimental design

no comments

Validity of the findings

no comments